# Morphological Study of Dental Structure in Dentinogenesis Imperfecta Type I with Scanning Electron Microscopy

**DOI:** 10.3390/healthcare10081453

**Published:** 2022-08-02

**Authors:** Andrea Martín-Vacas, Manuel Joaquín de Nova, Belén Sagastizabal, Álvaro Enrique García-Barbero, Vicente Vera-González

**Affiliations:** 1Department of Dental Clinical Specialties, Faculty of Dentistry, Complutense University of Madrid, 28040 Madrid, Spain; denova@ucm.es; 2Faculty of Dentistry, Alfonso X El Sabio University, 28691 Villanueva de la Canada, Spain; 3Pediatrician University Hospital of Getafe, 28905 Madrid, Spain; belen.sagastizabal@salud.madrid.org; 4Department of Conservative Dentistry and Prosthetics, Faculty of Dentistry, Complutense University of Madrid, 28040 Madrid, Spain; aegarcia@ucm.es (Á.E.G.-B.); vveragon@ucm.es (V.V.-G.)

**Keywords:** osteogenesis imperfecta, dentinogenesis imperfecta, dentin, dental enamel, tooth, deciduous, microscopy

## Abstract

*Background:* Dentinogenesis imperfecta type I (DGI-I) is a hereditary alteration of dentin associated with osteogenesis imperfecta (OI). *Aim:* To describe and study the morphological characteristics of DGI-I with scanning electron microscopy (SEM). *Material and methods:* Twenty-five teeth from 17 individuals diagnosed with OI and 30 control samples were studied with SEM at the level of the enamel, dentin–enamel junction (DEJ) and four levels of the dentin, studying its relationship with clinical–radiographic alterations. The variables were analysed using Fisher’s exact test, with a confidence level of 95% and asymptotic significance. *Results:* OI teeth showed alterations in the prismatic structure in 56%, interruption of the union in the enamel and dentin in 64% and alterations in the tubular structure in all of the cases. There is a relationship between the severity of OI and the morphological alteration of the dentin in the superficial (*p* = 0.019) and pulpar dentin (*p* 0.004) regions. *Conclusions*: Morphological alterations of the tooth structure are found in OI samples in the enamel, DEJ and dentin in all teeth regardless of the presence of clinical–radiographic alterations. Dentin structural anomalies and clinical dental alterations were observed more frequently in samples from subjects with a more severe phenotype of OI.

## 1. Introduction

Dentinogenesis imperfecta is a hereditary defect of dentin, affecting its structure and composition. In 1973 Shields et al. [1] proposed classifying hereditary dentin defects into two large groups, dentinal dysplasia and dentinogenesis imperfecta, with respective subgroups. Within dentinogenesis imperfecta there are three subgroups, type I (associated with osteogenesis imperfecta), type II (independent of osteogenesis imperfecta) and type III or Brandywine (identified in an isolated tri-racial population from southern Maryland and Washington D.C.). Dentinogenesis imperfecta (DGI) type I [1] is an alteration of dental development associated with osteogenesis imperfecta (OI) [2,3]. OI is a heterogeneous group of hereditary connective tissue disorders characterized by osteopenia, bone fragility and deformity [4,5] and a great tendency for fractures throughout life [6]. OI is considered a rare disease due to its low incidence, around 1:15,000 to 1:20,000 live births [7], without differences in terms of sex, race or ethnic group [5,7,8]. The etiopathological mechanism of OI is associated with qualitative and quantitative abnormalities of type I collagen, which in more than 90% of cases are caused by mutations in the genes that code for the alpha chains of collagen (COL1A1 and COL1A2 genes), while the remaining 10% is due to mutations that directly or indirectly interact with collagen during its metabolism [7,9,10,11,12,13].

DGI type I (DGI-I) is a very frequent finding in subjects with OI [14], affecting the primary dentition more than the permanent one [15,16]. The diagnosis of this dental affectation is mainly clinical and radiographic; alterations in dental colour, rapid dental attrition and detachment of enamel, bulbous crowns with accentuated cervical constriction and alterations in root development. However, the alterations are not always clinically detectable. Dental morphology in DGI-I is characterized by anomalies mainly at the dentin level, with irregularities in the tubular pattern and the presence of amorphous areas with calcification defects or vesicular inclusions; the involvement of the dentin–enamel junction (DEJ) is controversial since it can present without alterations or with an irregular or smooth scalloping [10,16,17]. Enamel is the least studied dental tissue in DGI-I, and there is no consensus about its alteration [17]. There are ultrastructural pathological findings in apparently healthy teeth [17,18], so it should be considered that all teeth could be affected to a greater or lesser extent. The large number of diagnostic criteria (clinical, radiographic and ultrastructural findings) make the diagnosis of dental involvement complex, making it difficult to make therapeutic decisions as there is no standardized dental treatment for subjects with this involvement.

Scanning electron microscopy (SEM) is an electron microscopy technique that provides images of the sample surface through the interactions of electrons and matter. Among its many advantages are the great depth of field, good image resolution and ease of sample preparation; however, it is an expensive technique and requires prior training. Only six previous studies have been found that analysed dental morphology with SEM in primary teeth [10,16,17,18,19,20,21], with sample sizes between 2 and 22 primary teeth. Enamel and DEJ have only been studied with SEM by three authors [10,17,20], observing fractures or defects in the mineralization of the enamel and damage to the DEJ scalloping and observing a flattening of the union between the enamel and the dentine [17,20], although its function was established to be normal [10]. Dentin is the most studied tissue with SEM [16,17,18,19,21], finding irregularities in the size and distribution of dentinal tubules, giant tubules, pulpal obliterations and non-calcified areas, among other findings.

Although over time various authors have studied dental ultrastructural anomalies in subjects with OI, the low incidence of the disease has made it difficult to carry out research with a large sample size, which would allow establishing a clear definition of dental morphological alteration in subjects with DGI-I. For this reason, we believe that a protocolized study with a large sample size is necessary to analyse the characteristics of dental involvement in subjects with OI, providing an adequate structural definition of DGI type I, to facilitate interprofessional communication and the clinical management of these patients. The aim of our research is to describe and study the dental affectation of primary teeth from subjects with OI at the level of the enamel, dentine–enamel junction and dentin with scanning electron microscopy.

## 2. Materials and Methods

This study has been supported by the AHUCE Foundation (Association of Crystal Bones of Spain) under the Collaboration Agreement with the Complutense University of Madrid. The study was conducted according to the guidelines of the Declaration of Helsinki and approved by the Ethics Committee of the Hospital Clínico San Carlos (17/326-E Thesis Code). The subjects and their parents or legal guardians were properly informed through written informed consent.

Primary teeth were selected according to inclusion and exclusion criteria (Table 1), obtaining a total sample of 25 teeth from 17 subjects, which were classified using the Sillence classification, following systemic clinical and radiographic criteria [22] (Table 2) in OI type I, III and IV.

All the extracted or exfoliated teeth of patients with OI belonging to the master’s degree in pediatric dentistry were collected, without calculating the sample size, due to the low prevalence of the disease. To establish a pattern of unaltered tooth structure, 30 primary teeth from healthy subjects were prepared and studied. Sample collection began in August 2017, ending in August 2018. The microscopic analysis of the samples was carried out from the initial date of sample collection until three months after the collection of the last study sample. The mean extraction age was 9.73 years for the study group and 10.14 years for the control group (Table 3).

The obtained teeth were preserved in a 35% formaldehyde solution reduced with distilled water in a one-quarter proportion. The teeth were sectioned longitudinally in the vestibular–lingual direction and processed by polishing with silicon carbide abrasive film discs (polishing machine 50-8435 MICRODUO-I AUTO) of decreasing abrasive grain (600 disc, 800/2400 disc and 1200/4000 disc) and subsequently treated with 37% orthophosphoric acid gel (Scotchbond ^TM^ etchant, 3M ^TM^) for 20 s to remove residues and impurities, and finally, metallized with gold in a vacuum chamber. For the ultrastructural study, a JEOL-JSM 6400 scanning electron microscope (SEM) (JEOL Ltd., Tokyo, Japan) was used.

The visualization and analysis of the samples was monitored with systematic observations at 500, 1000 and 2000 magnifications in enamel, dentin–enamel junction (DEJ) and dentin. The dentin observation was divided for observation into 4 regions of different depths: occlusal, middle, deep and pulpar (adjacent to the pulp chamber).

The morphological changes were classified into 4 categories (Figure 1) in order to be able to carry out correlations between the types of OI studied. The classification was created with the aim of standardizing the depth of observation in the samples. The presence of clinical or radiographic dental alterations and changes in tooth colour were recorded dichotomously. The severity of the clinical–radiographic alteration was recorded by the simultaneous presence of clinical–radiographic changes (severe), exclusively radiographic alteration (moderate) or the absence of clinical–radiographic pathology (mild).

A description of the sample was made by obtaining counts and percentages of the variables to be studied. Statistical analysis of the data was performed using Fisher’s exact test with a confidence level of 95% (*p* < 0.05) and asymptotic or bilateral significance. The Kappa–Cohen coefficient was carried out on 25 samples taken at random (45.45% of the sample) to evaluate the intra-operator agreement index, obtaining agreement values between good and excellent (Table 4).

## 3. Results

### 3.1. Ultrastructural Analysis of Enamel, Dentine–Enamel Junction and Dentin

Heterogenous findings were found in all dental structures: enamel, DEJ and dentin. In general terms, the alterations found could be described as loss of prismatic structure at the level of the enamel and tubular at the level of the dentin and interruption of the DEJ. The frequency and percentage of alterations in the study points can be seen in Table 5.

In the enamel, anomalies were found in 60% of the primary teeth of subjects with OI, in the form of interruption of the prisms and loss of the prismatic structure (Figure 1b) and inclusions of vesicles or ovoid formations inside (Figure 1c). Only in one tooth was the alteration severe, presenting an amorphous enamel, while in the remaining cases the alterations were between moderate and mild. Regarding the type of OI, the enamel alterations were very heterogeneous, with mild alteration being the most frequent in OI-I, while in OI-III and OI-IV the most frequent were moderate. The differences with the control group (Figure 1a) are statistically significant (*p* value < 0.001), but the differences between the OI types are not statistically relevant, although more severe enamel alterations are found in the more severe OI phenotypes (*p* value = 0.156).

In the DEJ at the superficial level, alterations were found in 64% of the primary teeth. In 28% of the sample, a partial fracture of the DEJ was observed, with alternation between regions with normal junction and altered areas; in 28%, a complete solution of continuity of the DEJ was observed with a clean separation between enamel and dentin (Figure 2b), without adherent remains of enamel; in 8% there is a total tissue separation including remains of enamel prisms (Figure 2c). The most frequent finding in OI-I was to find a partial separation of the DEJ, while in teeth with OI-IV it was to find a clean separation between enamel and dentin, and in teeth with OI-III it ranged between mild alteration and clean continuity solution. Clean rupture of the DEJ with both tissues separated was more frequent than the presence of a disruption with the remains of adhered enamel prisms in the three types of OI analysed. Despite the differences described, statistically no relationship was found between the alterations of the DEJ and the type of OI (*p* value = 0.956), although there were significant differences with respect to the control group (*p* value = 0.004) (Figure 2a).

The DEJ was analysed at the cervical level. In the same way as at the superficial level, the clean separation between enamel and dentin was the most frequent (44%), being more common in OI-IV and OI-III than in OI-I, and the alteration of the DEJ due to total fracture but adhered tissue remnants only occurred in one tooth of a patient with OI-I and another with OI-III. The data analysis showed that these differences in the involvement of the DEJ at the cervical level and the type of OI are not significant (*p* 0.351), although the differences with respect to the control group are significant (*p* < 0.001).

In the dentin, a generalized tissue disintegration and heterogeneity is observed with a decrease in the number of tubular dentin tubules, alteration of the diameter or tubular size, interglobular dentin and atubular areas (Figure 3). The appearance of the dentin was anomalous, with some samples showing dentin with a globular or cerebroid appearance. Significant differences were found in all the dentin points analysed compared to the control group (*p* < 0.001), with a higher frequency of alterations being observed in the OI group.

The superficial dentin region presented alterations in 40% of the study sample, being severe alterations with amorphous structure in 16% of the samples. Severe alterations were most frequent in OI-III, followed by OI-IV and absent in OI-I. The statistical analysis revealed that the different frequencies of involvement between the types of OI are significant (Figure 3a–d) (*p* value = 0.012).

The middle dentin was found to be altered in 100% of the primary dentition cases of subjects with OI, the moderate alteration being the most frequent with evident involvement of the tubular pattern. Severe alteration was more frequent in subjects with OI-III or OI-IV, and in OI-I it was absent. The statistical analysis determined that the differences regarding the type of OI are not significant (Figure 3e–h) (*p* value = 0.09).

The deep dentin was altered in 84% of the teeth with OI, being severe with an amorphous structure in only 32% of the cases. Severe alteration of deep dentin is more frequent in subjects with OI-III than in OI-I or OI-IV. The differences in deep dentin involvement based on the type of OI were not statistically significant (Figure 3i–l) (*p* value = 0.149).

The dentin adjacent to the pulp was found to be altered in 92% of the teeth with OI, being severe anomalies in 56% of the occasions, and less frequently moderate or mild. Severe anomalies are more frequent in OI-III than in OI-I or OI-IV, without finding this difference significant (Figure 3m–p and Figure 4) (*p* value = 0.249).

### 3.2. Relationship between Dental Alteration and Systemic Phenotype

The obtained teeth belonged to 17 subjects (35.29% OI-I, 41.18% OI-III and 23.53% OI-IV)), 94.1% treated with bisphosphonates from an early age. Dental alterations were found in 41.2% of the subjects, absent in OI-I and assuming 57.1% in OI-III and 75% in OI-IV (*p* value = 0.044). Radiographic alterations were manifested in 69.2% of the sample, present in 60% of OI-I, 66.7% of OI-III and 100% of OI-IV (*p* value = 1).

All the subjects with clinical alterations presented radiographic alterations; in addition, 55.6% of the subjects without clinical alterations presented alterations at the radiographic level (*p* value = 0.228). Ultrastructural alterations were present in all patients; 30.8% of the sample did not present clinical or radiographic alterations, while 38.5% presented radiographic alterations, and 30.8% simultaneously presented clinical and radiographic alterations, without significant differences in terms of the type of OI (*p* value = 0.364). Additionally, 75% of the subjects with clinical–radiographic alterations presented severe morphological dentin alteration, while 80% with radiographic alterations presented moderate morphological dentin damage; only 25% of the patients without clinical or radiographic alterations presented severe dentin alteration (*p* value = 0.172).

## 4. Discussion

The dental morphology of DGI-I is highly variable, finding ultrastructural anomalies that are fundamentally based on alterations in the tubular pattern and mineralization disorders. Alterations are found in the majority of teeth of subjects with OI, to a greater or lesser extent. These findings present a great variability in terms of their degree of severity. In this in vitro study, 25 teeth from 17 subjects with OI were systematically analysed with SEM, and a control group of 30 teeth was examined in order to establish dental structural normality. The SEM allows us to obtain a high magnification and resolution, providing 3D images of the dental surface. Comparison with previous studies [16,17,18,19,20,24,25,26,27,28,29,30,31,32,33] is difficult since most of them are purely descriptive; in addition, the sample size is very uneven and the type of microscopy varied between studies. Due to the low incidence of DGI-I, we believe that our research can provide important data about the dental morphology of teeth with DGI-I and its relationship with systemic disease.

In previous investigations in subjects with OI, no histological alterations in the enamel were found [10,17,29] and the tendency to fracture or detachment of the enamel is justified by the loss of the DEJ scalloping and the involvement of the underlying dentin. Despite this, Lindau [20] described alterations in the mineralization of the primary enamel, especially in cases of clinical DGI, although without finding differences between the different types of OI. On the other hand, some authors [30,33] established that, although the enamel is apparently normal, there are areas with fractured enamel prisms and an irregular trend of the enamel prisms and unpacked crystals of hydroxyapatite in all teeth. We found alterations corresponding to loss of crystalline structure, interruption of the lamellar pattern and amorphous regions of variable severity in 60% of primary teeth. In those teeth in which the anomaly corresponds to fractures of the enamel prisms, we cannot establish whether these have been caused by being supported by an altered dentin, or by a specific affectation of the enamel. Coinciding with Lindau [20], with our results we cannot establish a relationship between the systemic phenotype of OI and the alteration of dental enamel.

There is controversy about the involvement of the DEJ, because in some studies it is stated that the DEJ has a normal structure and function [10,16,17,29], while some [17,20] state that it is altered. Lindau and collaborators [20] describe the alternation of pathological and normal regions. They also established that the normal, scalloped appearance is more frequent in the cervical third of the DEJ. In our sample, in 28% of primary teeth a complete separation of the DEJ was observed, with the most common finding being a fracture of the union in which the enamel and dentin were separated and less common a fracture of the joint in which there are traces of enamel adhered to the dentin, and therefore, we assume that it is due to faults in the enamel and not only in the DEJ. In 28%, the alterations found were very heterogeneous, with partial fractures of the DEJ being observed and altered and non-altered areas coexisting. The failure of enamel and dentin to cross-link can lead to poorer mechanical retention between both tissues and therefore to functional failure of the DEJ. This retentive failure of the enamel on the dentin would advantage the detachment of the dental enamel.

Dentin has been the most studied dental structure in DGI-I and therefore the one in which most signs of structural anomaly have been described. Regarding the aetiology of dentin alterations, Majorana et al. [16] established that dentin anomalies are possibly a consequence of odontoblast dysfunction, and Lygidakis et al. [17] hypothesized that tubular obliteration is due to mineralization. Multiple investigations have described ultrastructural findings of dentin [16,17,19,24,25,26,29,31,32,33], which can be summarized in the presence of irregular dentinal tubules with alterations in their direction and the presence of both giant and obliterated tubules with circular areas of intertubular matrix without calcification. Similar ovoid areas embedded in dentin have been observed in studies in type II DGI, but it is unclear whether they are unmineralized areas or giant or irregular tubules that have fused together [34]. In addition, the alternation of areas of apparently normal dentin and regions with abnormal dentin has been described. These findings are consistent with our results, in which we found very similar images of varying severity in the four dentin points examined, corresponding to the presence of dentin tubules with a very heterogeneous pattern, with changes in tubular diameter, reduction in tubular density with the presence of atubular regions or with remnants of obliterated tubules, presence of giant tubules and, in the most severe cases, images of totally dysplastic, amorphous and atubular dentin with a globular or cerebroid appearance, which is similar to the findings previously described by other authors.

Malmgren and Lindskog [18] created a dysplastic dentin scale through clinical–radiographic assessment; their findings indicated that the dentin presents greater dysplasia in the circumpulpar area than in the mantle dentin. In addition, they did not find differences in affectation between the primary and permanent dentition. De Coster [28] and Hall et al. [10] analysed the dentin at different depth points, establishing that the mantle dentin and the first layer of tubules is apparently normal, ending abruptly in a zone parallel to the UAD in the which obliterated tubules are found and from which normal and dysplastic or atubular areas alternate. Our data indicate that all the teeth show dentin involvement in at least one point of their extension. Regarding our results, based on the depth of analysis, the dentin closest to the DEJ is the least frequently altered in subjects with OI (40%), being followed by a layer of dysplastic dentin in all cases, leaving passage to the deep and pulpar dentin altered in 84% and 92% of cases, respectively.

Type I collagen is altered both systemically and dentally; therefore, a correlation could be established regarding the severity of the dysplasia. However, collagen does not have the same function in bone and dentin, so the severity of bone and dentin involvement is highly variable [15]. Some authors established that there are no dental histological differences between the different types of OI [16], while others [18,28] established that there are morphological differences in dentin, with more frequent and marked findings in patients with OI type III and IV than in type I. Malmgren and Lindskog [18], despite observing that the manifestations of dysplasia in the mantle dentin increase with the severity of the OI, did not find a relationship between the type of OI and dysplasia dentin. Derived from our results, we can affirm that, in general, more severe dental morphological anomalies are found in subjects with a more severe phenotype of the disease. By differentiating four regions of different depths of dentin, we found a significant relationship between dental and systemic severity in superficial dentin, although in all the areas studied anomalies were found more frequently in subjects with more severe OI.

Malmgrem [35] classified subjects with OI into four groups based on the severity of dental involvement and the presence of dental agenesis, finding severe alterations in 47.46% of the participants and finding that dental involvement was more severe in the phenotypes of more severe OI. Malmgren and Lindskog [18] related two semi-quantitative classification systems, the clinical–radiographic scale and the dysplastic dentin scale, finding a relationship between both scales. The dysplastic dentin scale proves the existence of subclinical morphological alterations in teeth apparently without DGI-I. Taqui et al. state that pulpar discoloration and obliteration are more frequent in phenotypes III and IV, but establish that the analysis of genetic mutations is a better predictor of the dental phenotype than the type of OI [36]. Xi et al. also state that the clinical manifestation of DGI-I is more frequent in the most severe phenotypes of systemic disease [37]. In our study, we found morphological alterations in all the subjects analysed, even in the absence of clinical–radiographic pathology. Recent studies affirm that there are changes in molecular and physical characteristics of the dentin of patients with OI even in the absence of manifest DGI-I [38]. Although the alterations were more frequent in subjects with a systemic phenotype of severe OI, the differences were not significant, except for dental clinical manifestations, since it was absent in subjects with OI-I. In addition, abnormal radiographic findings were observed in all clinically abnormal teeth. The severity of the radiographic involvement was attempted with the presence of clinical and radiographic signs, observing that in 75% of the subjects with simultaneous clinical and radiographic alterations the morphological involvement was severe, assuming only 25% of the subjects without clinical and radiographic alterations are compatible with DGI-I.

Regarding the strengths, our research presents a complete study, including dental morphology, and the clinical–radiographic examination of the patient, which allows us to consider that all the patient’s teeth present subclinical morphological alterations, although clinically or radiographically they do not present findings compatible with DGI-I. This reflection is of vital importance when planning the treatment of the patient and advising the families. In previous studies that analyse the dental ultrastructure in patients with OI, the sample size ranges from 1 to 22 primary teeth and 1 to 11 permanent teeth, so our sample size exceeded those previously studied. It would be interesting to be able to increase the sample size in order to determine the interrelation between alterations at different depth levels as well as to study the influence of genetic mutation and medication in the presence of dental alterations associated with OI. 

In addition, in recent decades, genetic studies have been carried out that have allowed a better understanding of both the etiopathogenesis of OI and its relationship with the systemic and dental phenotypes [32,36,37,38,39,40,41,42,43]. Secondary to the early diagnosis of current OI, often even during pregnancy, the vast majority of these subjects have received pharmacological treatment from birth, usually with intravenous bisphosphonates, so it is not surprising that a relationship is found between the period or dose of administration of these drugs and the radiographic results of DGI-I [44]. 

Our study has some limitations, such as the small sample size in relation to the low prevalence of systemic disease or the lack of genetic reports for the patients or the cumulative dose of bisphosphonates of the patients at the time of dental exfoliation or extraction, which would enrich the results and increase the knowledge of the pathophysiology of the dental morphological alteration. Despite the aforementioned limitations, we believe that our research may have great clinical relevance by showing that the dental ultrastructural morphological alteration can occur despite not clinically manifesting DGI-I. This allows us to better advise the patient when it comes to preventing possible dental problems such as fractures or susceptibility to caries, in addition to taking special care in the selection of our dental adhesive materials.

Since the classification of clinical severity of OI brings together patients with different genetic mutations and an individualized schedule of drug administration, it would be interesting in future investigations to know the complete genetic analysis and medication administration schedule (time and route of administration, average accumulated dose, etc.) of patients with OI in order to be able to correlate the dental ultrastructure of these subjects and their clinical manifestation.

## 5. Conclusions

Teeth with DGI-I present morphological alterations of the dental structure in the enamel (64%), UAD (64–72%) and dentin (100%). Enamel alterations include fractures of the enamel prisms and progressive loss of the prismatic pattern. Involvement in the RAU can present as a clean continuity solution between enamel and dentin, or with remains of enamel prisms adhered to dentin. Dentin alterations are characterized by a decrease in the diameter and number of dentin tubules, with an increase in atubular areas, which can manifest with a completely dysmorphic appearance. The findings indicate that there is a relationship between OI and dentin involvement in the superficial dentin region, with greater dysmorphic findings found in subjects with a more severe phenotype of systemic disease. Clinical signs of DGI-I are present in 41.2% of the subjects and radiographic signs in 69.2%. The clinical alteration is significantly higher in subjects with severe OI phenotypes. The ultrastructural morphological alteration occurs in all teeth, regardless of the clinical and/or radiographic manifestations.

## Data Availability

The data presented in this study are available on request from the corresponding author.

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
