# Peer review of "Morphological Study of Dental Structure in Dentinogenesis Imperfecta Type I with Scanning Electron Microscopy"

_healthcare, 2022, doi:10.3390/healthcare10081453_

Round 1
Reviewer 1 Report
The manuscript under review attempts to evaluate the morphology of Dental Structure in Dentinogenesis Imperfecta Type I using Scanning Electron Microscopy, and the manuscript did capture the details of the study design, methodology, and implementation of the project. All the sections of the manuscript are well written, but they need several corrections. Kindly find attached the manuscript’s detailed comments and suggestions below in all the sections, which will help the authors check and revise the manuscript.
Title: kindly correct Microscopy
Abstract:
1. Kindly provide structured abstract: Mention subheadings Background, Aim
2. Kindly provide Mesh keywords
Introduction:
1. Write briefly about dental structure morphology in DI
2. Write briefly about SEM
3. Literature on SEM and DI
Materials and Methods:
1. Provide ethical approval details at the start of the methodology
2. Provide a copy of the informed consent
3. Table 1: correct de with the
4. Table 2: kindly provide a complete form of AD and AR as a table footer.
5. Did the authors conduct sample size calculations? Kindly provide details
6. Included samples enough to justify?
7. Did the authors extract the teeth from the patients? Or the teeth extracted for the research purpose?
8. diamond disks: Mention details (manufacturer, number, city and country)
9. 37% orthophosphoric acid gel: Mention details (manufacturer, number, city and country).
10. JEOL-JSM 6400 scanning electron microscope: Mention details (manufacturer, number, city and country)
11. The morphological changes were classified into 4 categories: References?
12. Was the examiner calibrated?
13. Provide a table for the Kappa-Cohen coefficient
Results:
1. Kindly cite Figures 1a and 2a in the text
2. Figures need a scale bar included
3. Provide each image with different magnifications (at least 4 magnification)
4. Table 5 is big and confusing, kindly try to split it and make it reader easy
5. Avoid repetition of results in text, just mention the significant difference rest all can be seen in the tables
Discussion:
1. 2nd paragraph is out of context, lines 249 to 258- advised to delete
2. Write the limitations of the current study
3. Provide Clinical importance and future directions
Author Response
Dear #1 reviewer. Thank you very much for your time and willingness to review our manuscript. We have modified the manuscript (marked in red) and have answered some of your considerations, which we hope you find correct.
- Sorry for the typo in the title, it's fixed now.
- Subtitles in italics have already been added to the abstract to make it more structured, as well as introducing line breaks between them.
- Keywords have been written using Mesh terms.
- In the introduction we have added a brief introduction to dentinogenesis imperfecta, electron microscopy and the background in the use of scanning electron microscopy in the analysis of dental ultrastructure in patients with osteogenesis imperfecta.
- We have provided the ethical approval details at the start of the methodology.
- We have not added de informed consent as a appendix because it is written in Spanish, but we have attached to you in other information in the platform.
- We have corrected typos and table footers.
- In relation to the sample size calculation, no, the sample size was not calculated since ALL patients with OI from the Master's Degree in Paediatric Dentistry were selected. The decision not to perform a sample size calculation was based on the low prevalence of this rare disease. This consideration has been added in material and methods.
- Yes, since as I have mentioned, the prevalence of OI is very low, and because of that we think the sample is enough to justify the results. In addition, previous studies with scanning electron microscopy have a sample size that ranges between 1 and 22 primary teeth, so as our sample is superior, we consider that it improves the previous evidence. It is proceeded to add in the paragraph of limitations and strengths of the study.
- As specified in Table 1, the teeth obtained are teeth with spontaneous exfoliation donated by the patients or teeth extracted for a justified reason, unrelated to the study. The most common causes are caries or orthodontic reasons. Teeth were not extracted exclusively for research purposes, a subject that would never have been approved by the ethics committee.
- Added the model of polishing machine used, the hardness of the polishing discs and the data of the orthophosphoric acid gel and the SEM machine.
- It is our classification, it is not referenced because that classification was created for the researchers to categorize the severity of the structural damage.
- Yes, of course the examiner was calibrated, as we stated in the last part of material and methods. The main examiner is a pediatric dentist and an expert on the subject, so she was trained and calibrated, also establishing an intraoperative concordance index. We proceed to add the Kappa-Cohen index table.
- We have cited figures 1a and 2a in the text
- A scale bar has been added in the figures
- For reasons of space and ease of reading, different magnifications of each image have not been added. The authors and I would like to know if you consider it essential, or if you want us to provide this information on a specific section.
- Table 5 have been modified in order to improve its quality and comprehension
- The results have been corrected in order not to repeat information
- The paraparaph you stated was out of context has been deleted
- We have written the limitations and strengths of the study, and the clinical importance and future directions.
We are at your disposal if you need more information about our manuscript.
Sincerely,
Dr. Andrea Martín Vacas.
Reviewer 2 Report
This study studied the morphological characteristics of DGI-I with scanning electron microscopy (SEM). They found morphological alterations of the tooth structure are found in OI samples in the enamel, DEJ and dentin in all teeth regardless of clinical-radiographic alterations. In addition, the dentin structural anomalies and clinical dental alterations were observed more frequently in samples from subjects with a more severe phenotype of OI. The research is explained well and discussed. Some comments.
Introduction.
It is better to give an introduction to the DI and then explain the types.
Materials and Method
It is better to add a radiological, photographs, or diagrammatic sketch regarding the coding of structural morphological changes.
Please mention the start and end of the experiment date.
Results
In Figure1 and 2, the scales are not visible. It is better to add a better scale.
Add sub-labeling in Figure 2 in a better way.
Line 86. eh is not clear.
Figure 3. Add the explanation of the Figure below the Figure.
Author Response
Dear #2 reviewer,
Thank you very much for your dedication to our manuscript. We have proceeded to make changes (marked in red) and to respond to your considerations.
A brief introduction to hereditary defects of dentin is added to begin the introduction “Dentinogenesis Imperfecta is a hereditary defect of dentin, affecting its structure and composition. In 1973 Shields et al. [1] proposed to classify hereditary dentin defects into two large groups, Dentinal Dysplasia and Dentinogenesis Imperfecta, with respective subgroups. Within Dentinogenesis Imperfecta there are three subgroups, type I (associated with Osteogenesis Imperfecta), type II (independent of Osteogenesis Imperfecta) and type III or Brandywine (identified in an isolated tri-racial population from southern Maryland and Washington D.C.).”
I understand your consideration, in order to improve the understanding of the coding. We have created a scheme to replace the coding table and make it easier to read.
We have mentioned the start and end date of the research.
The line and scale reading have been enlarged to be visible on all figures.
The sublabeling of figure 2 has been improved, as well as the explanation of it.
Line 86 belongs to table 3 and in my version no error appears. If you still see it in the new version please let me know and we will proceed to consider the change. If you are referring to the last row reading, we have added "Total" to specify that it is the total number of teeth in the control sample.
Following your instructions, we have proceeded to put the explanation of figure 3 later and not at the beginning as we had done.
We are at your disposal if you need more information about our manuscript.
Sincerely,
Dr. Andrea Martín Vacas.
Round 2
Reviewer 1 Report
Dear Authors,
The authors have addressed all the comments and the manuscript is much improved and gained the status of acceptance. I suggest accepting the manuscript in its present form. I would like to congratulate the authors and appreciate their hard work and wish them all the very best in future endeavours.
Best regards and keep well